# D²-Former: Mixture-Of-Experts Guided Dual Transformer for Multi-Scale Medical Image Segmentation

**Md Sohag Mia**[*1,2]                  SHUVO2018@NUIST.EDU.CN
**Aya Taourirte**[*1]                 202551620018@NUIST.EDU.CN
**Muhammad Abdullah Adnan**[2]            ADNAN@CSE.BUET.AC.BD
**Wenlong Ming**[†1]                  WMING@NUIST.EDU.CN

[1] *School of AI, Nanjing University of Information Science and Technology, Nanjing, China*

[2] *School of Computer Science, Bangladesh University of Engineering and Technology, Dhaka, Bangladesh*

**Editors:** Accepted for publication at MIDL 2026

## Abstract

Precise delineation of anatomical structures from medical images is critical for clinical diagnosis and treatment planning, yet remains profoundly challenging due to ambiguous boundaries, extreme scale variations, and the heterogeneous appearances of pathological tissues. Current segmentation methods frequently fall short in effectively balancing global contextual understanding with adaptive, multi-scale feature fusion, limiting their robustness across diverse clinical scenarios. To address these limitations, we propose D²-Former, a novel encoder-decoder framework that integrates a dual-encoder architecture–combining a Swin Transformer for hierarchical local-global modeling and a DINOv3 foundation model for high-fidelity dense feature extraction—with a Softer Mixture-of-Experts (Softer-MoE) module for input-adaptive feature refinement. Our design further introduces a Spatial-Frequency Gated Channel Attention (SF-GCA) module to fuse complementary encoder representations and a Residual Attention Decoder (RAD) with deep supervision for progressive map reconstruction. Extensive experiments across nine public benchmarks–spanning polyp segmentation, retinal vessel delineation, multi-organ abdominal CT segmentation, and nuclei instance segmentation–demonstrate that D²-Former achieves state-of-the-art or highly competitive performance. The model exhibits strong generalization across varied anatomical scales, imaging modalities, and clinical scenarios, underscoring its potential for reliable computer-assisted diagnosis.

**Keywords:** DINOv3, Medical Image Segmentation, Mixture-of-Experts

## 1. Introduction

Medical image segmentation is crucial for computer-assisted diagnosis and treatment yet remains challenging due to low contrast, ambiguous boundaries, and anatomical variations (Rizhi et al., 2025; Ou et al., 2022; Zan et al., 2023; Hatamizadeh et al., 2022; Sharif et al., 2022). Convolutional networks like U-Net (Ronneberger et al., 2015) capture local features but miss long-range dependencies (Hatamizadeh et al., 2022), while transformers like Swin-UNet (Cao et al., 2022b) model global context but lack spatial biases for fine details. Hybrid models with static fusion struggle with medical images' heterogeneous scales and textures.

---

[*] Contributed equally

[†] Corresponding author

Mixture-of-Experts (MoE) architectures dynamically route information to specialized experts, enabling adaptive capacity without proportional computation (Lepikhin et al., 2020; Riquelme et al., 2021). Softer-MoE (Puigcerver et al., 2024) further allows soft expert combination, producing representations suited for ambiguous boundaries. Meanwhile, self-supervised foundation models like DINOv3 offer high-fidelity features for segmentation (Gao et al., 2025; Yang et al., 2025), but their frozen backbones or simple fusion limit adaptation to medical domains. Related to our goal of complementary encoders, DSU-Net combines DINOv2-guided feature collaboration with a SAM2 backbone using lightweight adapters and cross-layer fusion while largely freezing the foundation backbones (Xu et al., 2025). SAM2-UNeXT (Xiong et al., 2025b) further extends SAM2-UNet (Xiong et al., 2026) by integrating an auxiliary DINOv2 (Oquab et al., 2023) encoder with a dual-resolution strategy and a dense glue layer for encoder fusion. In contrast, $D^2$-Former does not adapt a SAM/SAM2 promptable segmentation pipeline; instead, we couple a hierarchical Swin encoder with a DINOv3 encoder and introduce (i) Softer-MoE inside Swin for input-adaptive refinement and (ii) SF-GCA for stage-wise gated fusion tailored to multi-scale medical segmentation.

To address these gaps, we propose $D^2$-Former, a Mixture-of-Experts guided dual Transformer framework that couples a Swin Transformer encoder with a DINOv3 foundation-model encoder. A Spatial-Frequency Gated Channel Attention (SF-GCA) module aligns and fuses DINOv3's high-fidelity dense features with Swin's hierarchical representations, while Softer-MoE enables adaptive feature refinement within the Swin branch. Unlike prior DINO-based segmentation or MoE vision transformers, $D^2$-Former couples a Swin encoder and a DINOv3 foundation encoder via SF-GCA and Softer-MoE specifically for medical image segmentation. The contributions of this work are threefold. First, we introduce a dual-encoder architecture that combines Swin and DINOv3, providing complementary hierarchical local–global modeling and rich semantic feature extraction suitable for diverse medical imaging modalities. Second, we embed Softer-MoE into Swin Transformer blocks, replacing static feed-forward layers with input-dependent expert mixing to better handle large anatomical scale variations and ambiguous boundaries. Third, we design SF-GCA and a Residual Attention Decoder (RAD) with Squeeze and Channel Excitation (CSE), Squeeze and Spatial Excitation (SSE) (Fitzgerald et al., 2024) and deep supervision to adaptively fuse dual-encoder features and progressively recover fine-grained segmentation maps. Extensive experiments across polyp, retinal vessel, multi-organ abdominal CT, and nuclei segmentation benchmarks demonstrate that $D^2$-Former achieves improved generalization and consistently enhanced segmentation quality under challenging appearance variations.

## 2. Related Work

Transformers have significantly advanced medical image segmentation by modeling long-range relationships. Early approaches such as TransUNet (Chen et al., 2021a) combined ViT with U-Net, while UNETR (Hatamizadeh et al., 2022) used a pure Transformer encoder for volumetric prediction. Later architectures such as Swin-UNet (Cao et al., 2022b) and PVT (Wang et al., 2021) introduced hierarchical attention for scalable multi-scale feature extraction. Hybrid and dual-path methods (e.g., MedFuseNet (Chen et al., 2025), DTA-SUnet (Ma et al., 2024), GLFNet (Sun et al., 2024)) mitigate Transformers' limited spatial

priors via multi-branch fusion, but typically rely on static fusion that may under-adapt to large appearance and scale variations.

Foundation models further improve transferability. Self-supervised DINOv3 (Siméoni et al., 2025) provides high-fidelity dense representations; DINO U-Net (Gao et al., 2025) uses DINOv3 as a frozen encoder with adapters, and SegDINO (Yang et al., 2025) employs a lightweight MLP decoder, both demonstrating DINOv3's transfer learning superiority (Wang et al., 2025). However, they either lack domain adaptation or fail to handle pathological scale variability. In parallel, promptable "segment anything" models have been customized for medical segmentation: SAMed (Zhang and Liu, 2023) adapts SAM via LoRA-based fine-tuning of the image encoder together with the prompt encoder and mask decoder, while SAMed-2 (Yan et al., 2025) builds on SAM-2 (Ravi et al., 2024) by introducing a temporal adapter and a confidence-driven memory mechanism for robust multi-task medical segmentation. Recent SAM2-UNet (Xiong et al., 2026) shows SAM2's Hiera backbone can serve as a strong encoder for U-shaped models with parameter-efficient adapters, and SAM3-UNet (Xiong et al., 2025a) extends this idea to SAM3 (Carion et al., 2025) using a simple adapter and lightweight U-Net-style decoder for low-cost downstream adaptation.

Mixture-of-Experts (MoE) improves feature specialization and capacity. Sparse MoE (Lepikhin et al., 2020; Riquelme et al., 2021) routes tokens dynamically for efficient scaling. Vision adaptations such as Patcher (Ou et al., 2022) and M$^3$ViT (Liang et al., 2022) show MoE gating enhances representation diversity, and Softer-MoE (Puigcerver et al., 2024) softly combines experts for stable representations suitable for ambiguous boundaries and variable scales. These advances motivate integrating Softer-MoE into our dual-encoder framework with SF-GCA for adaptive multi-scale medical segmentation.

## 3. Methods

Our D$^2$-Former architecture, illustrated in Figure 1, integrates a DINOv3 and Swin Transformer Module for multi-scale context and our novel Softer-MoE for dynamic feature specialization, which are subsequently fused via SF-GCA modules before being decoded through a U-Net-style decoder with squeeze-and-excitation attention. Below we describe our network's details.

### 3.1. Dual Transformer Module

The Dual Transformer Module integrates a Swin Transformer (Base) and a ViT-S+/16 DINOv3 encoder to provide complementary multi-scale features. DINOv3, a self-supervised foundation model, yields high-quality dense representations through discriminative SSL and Gram-anchored training. DINOv3's first 6 blocks are frozen during training to preserve generic low-level representations and reduce overfitting on medical datasets; higher blocks remain trainable for domain adaptation.

*Swin branch.* Given an image $X \in \mathbb{R}^{H \times W \times C}$, non-overlapping $s_1$-sized patches form tokens $X_1 = \text{PatchEmb}(X, s_1)$, with $X_1 \in \mathbb{R}^{\frac{H}{s_1} \times \frac{W}{s_1} \times C_1}$. These pass through hierarchical Swin stages to produce multi-scale features $F_1 = \{F_s^i\}_{i=1}^4$ (after each patch merging).

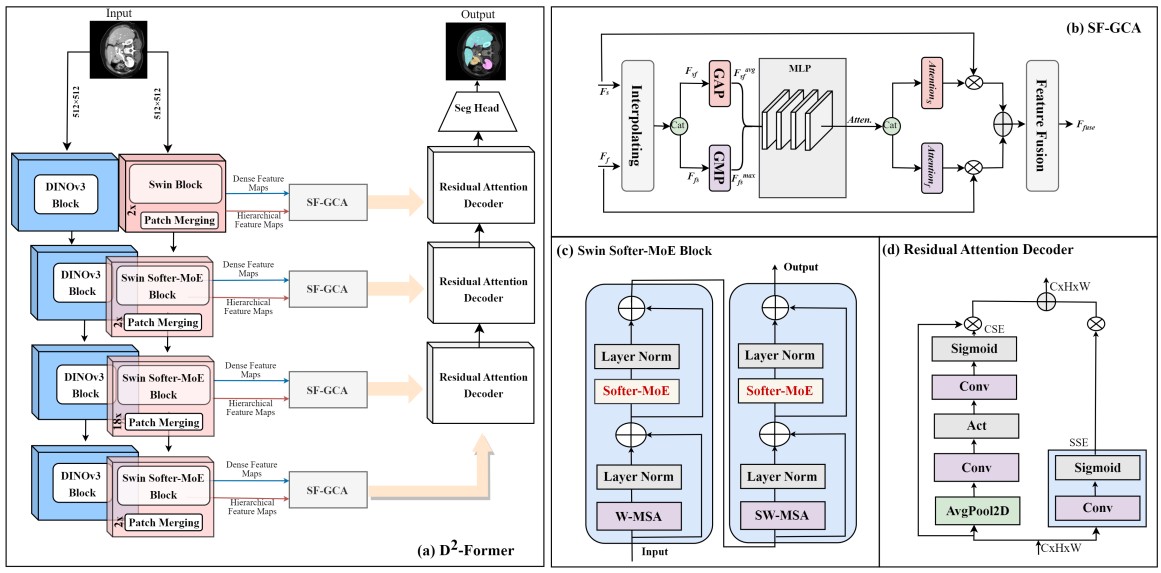

Figure 1: The architecture of our proposed D$^2$-Former model. The '18×' indicates that stage-3 has depth 18 in the chosen Swin configuration (depths 2, 2, 18, 2); hence, the corresponding block repeats 18 times. In (b), 'Cat' denotes channel-wise concatenation of the aligned features $[F_s; \hat{F}_f]$.

*DINOv3 branch.* Following the ViT formulation, the image is patchified with size $s_2$, yielding $Z^{(0)} \in \mathbb{R}^{N \times d}$ where $N = \frac{H}{s_2} \cdot \frac{W}{s_2}$. The ViT-S backbone applies $L$ Transformer blocks,

$$Z^{(\ell)} = B_\ell(Z^{(\ell-1)}), \quad \ell = 1, \ldots, L, \tag{1}$$

and we extract token features from four layers $\{\ell_i\}_{i=1}^4 \subset \{1, \ldots, L\}$, yielding the multi-level DINOv3 representation $F_2 = \{Z^{(\ell_i)}\}_{i=1}^4$. In practice we set $\{\ell_i\} = \{3, 6, 9, 12\}$ to align the four DINOv3 levels with the four Swin stages. Each $Z^{(\ell_i)}$ is reshaped into a 2D feature map, denoted $F_f^i$ when used in SF-GCA (Sec. 3.3).

*Fusion and decoding.* At each stage $i$, we fuse the Swin feature $F_s^i$ with the corresponding DINOv3 feature map $F_f^i$ using SF-GCA, which performs channel projection and spatial alignment (Eq. (7)) followed by gated fusion:

$$F_{\text{out}}^i = \text{SF-GCA}_i(F_s^i, F_f^i), \quad i = 1, \ldots, 4. \tag{2}$$

The fused $\{F_{\text{out}}^i\}_{i=1}^4$ serve as skip features for the residual attention decoder to restore full-resolution segmentation.

*Why dual encoders help?* The dual encoder combines Swin's strength in modeling hierarchical structures with DINOv3's robustness in semantic feature extraction. Let $Y$ be the target mask and $(F_s^i, F_f^i)$ the stage-$i$ features from Swin and DINOv3. By the chain rule, $I(Y; F_s^i, F_f^i) = I(Y; F_s^i) + I(Y; F_f^i \mid F_s^i) \geq I(Y; F_s^i)$, with strict gain when $F_f^i$ provides complementary cues. SF-GCA preserves this conditional gain via alignment and channel-wise gating, allowing the decoder to exploit non-redundant multi-scale information.

### 3.2. Swin Softer-MoE Block

We integrate Softer-MoE into Swin Transformer blocks to replace static FFNs with input-adaptive expert mixing. Softer-MoE uses learnable convex (Puigcerver et al., 2024) combinations of experts, enabling content-dependent refinement with minimal computational overhead. We start Softer-MoE from stages 2–4 to avoid the $O(m^2)$ memory cost at stage 1, because stage 1 has the largest token count, making it memory/compute inefficient and less relevant to semantic scale variation. We emphasize stage 3 (depth 18) since it is the main mid-resolution representation where multi-scale anatomy and boundary ambiguity are most pronounced. We use 8 experts because it provides the best accuracy/compute trade-off in our ablation (Table 7), and soft dense mixing mitigates expert collapse without an extra load-balancing loss.

*Slot Construction and Routing.* Given tokens $X \in \mathbb{R}^{m \times d}$, each layer uses $n$ experts with $s = m/n$ slots. Slots $\tilde{X}$ are computed via dispatch weights D

$$D_{ij} = \frac{\exp((X\Phi)_{ij})}{\sum_{i'=1}^{m} \exp((X\Phi)_{i'j})}, \qquad \tilde{X} = D^{\top}X, \tag{3}$$

where $\Phi \in \mathbb{R}^{d \times (ns)}$ is learnable. The column-wise softmax encourages each slot to aggregate information from all tokens.

*Expert Processing and Output Combination.* Each slot $\tilde{X}_i$ is processed by expert $f_{\lfloor i/s \rfloor}$. The outputs $\tilde{Y}$ are merged using combine weights C and it's the result of applying softmax over the rows of $X\Phi$

$$C_{ij} = \frac{\exp((X\Phi)_{ij})}{\sum_{j'=1}^{ns} \exp((X\Phi)_{ij'})}, \qquad Y = C\tilde{Y}. \tag{4}$$

*Stage Integration.* Following Swin's hierarchical reduction $m/2^{3(i-1)}$, Softer-MoE is inserted at stages 2–4. We use $n = 8$ experts and place Softer-MoE layers at (0, 3, 5, 7, 9, 12, 15, 17) in stage 3, and (0, 1) in stages 2 and 4. Stage-3 contains 18 Swin blocks, so we place more Softer-MoE layers there than in stages 2 and 4 (each depth 2). We do not add an explicit load-balancing loss; with 8 experts and dense soft routing, we did not observe expert collapse or training instability.

*Swin Softer-MoE Block.* Each block alternates window-based attention and Softer-MoE:

$$\hat{z}_l = W\text{-}MSA(LN(z_{l-1})) + z_{l-1}, \qquad z_l = Softer\text{-}MoE(LN(\hat{z}_l)) + \hat{z}_l, \tag{5}$$

$$\hat{z}_{l+1} = SW\text{-}MSA(LN(z_l)) + z_l, \qquad z_{l+1} = Softer\text{-}MoE(LN(\hat{z}_{l+1})) + \hat{z}_{l+1}. \tag{6}$$

This preserves Swin's local–global hierarchy while enabling dynamic, token-dependent expert routing. The Swin Softer-MoE block is depicted in Figure 1(c).

### 3.3. Spatial-Frequency Gated Channel Attention

We introduce a Spatial-Frequency Gated Channel Attention (SF-GCA) module to fuse the complementary representations from the dual encoders (Figure 1(b)). At encoder stage $i$, the Swin encoder yields a spatial feature map $F_s^i \in \mathbb{R}^{C_s \times H \times W}$, while DINOv3 provides a globally contextual frequency-rich feature map $F_f^i \in \mathbb{R}^{C_f \times H' \times W'}$. For readability, we omit the stage index $i$ below.

To match dimensions, $F_f$ is projected from $C_f$ to $C_s$ channels via a $1 \times 1$ convolution, followed by GroupNorm and spatial alignment:

$$\hat{F}_f = \text{Align}(\text{GN}(\text{Conv}_{1 \times 1}(F_f))). \tag{7}$$

Here Align$(\cdot)$ denotes 2D bilinear interpolation to match the spatial resolution of $F_s$. The aligned features are concatenated and processed through a dual-branch channel attention mechanism using global average pooling (GAP) and global max pooling (GMP). Both descriptors pass through a shared two-layer MLP (reduction ratio $r = 16$) with sigmoid activation:

$$A = \sigma\Big(\text{MLP}(\text{GAP}([F_s; \hat{F}_f])) + \text{MLP}(\text{GMP}([F_s; \hat{F}_f]))\Big). \tag{8}$$

Here $[\,;\,]$ denotes channel-wise concatenation. Since $[F_s; \hat{F}_f] \in \mathbb{R}^{2C_s \times H \times W}$, the attention vector satisfies $A \in \mathbb{R}^{2C_s}$. We split it as $A = [A_s; A_f]$ with $A_s, A_f \in \mathbb{R}^{C_s}$ (first $C_s$ and last $C_s$ ($C_f$) channels), and apply channel-wise gating: $\tilde{F}_s = A_s \odot F_s$ and $\tilde{F}_f = A_f \odot \hat{F}_f$, where $\odot$ denotes per-channel multiplication. Fusion is then performed through learnable adaptive weighting:

$$F_{fused} = \frac{\alpha \tilde{F}_s + \beta \tilde{F}_f}{\alpha + \beta}, \tag{9}$$

where $\alpha$ and $\beta$ are trainable scalars. This enables dynamic balancing between spatial detail and frequency-domain context, improving cross-encoder consistency across scales. We instantiate SF-GCA at all four encoder stages so that fused features are available at each scale for the decoder.

### 3.4. Residual Attention Decoder

We use a Residual Attention Decoder (RAD) (Figure 1(d)) to progressively reconstruct dense segmentation maps. Each stage upsamples the incoming feature map to the skip resolution, concatenates the encoder shortcut (when present), and applies a Spatial and Channel Squeeze and Excitation (SCSE) module to recalibrate channel and spatial responses. Two consecutive $3 \times 3$ layers follow for local aggregation, after which a second SCSE refines the output. The final segmentation head applies a Residual Block (RB) with Mish activation and GroupNorm for last-stage refinement prior to upsampling and $1 \times 1$ prediction. The RB is therefore a post-decoder refinement module.

During training, we employ deep supervision with two auxiliary output branches connected to intermediate decoder stages. The auxiliary losses are weighted by factors 0.2 and 0.1 respectively (reduced to 0.1 and 0.05 during warmup), encouraging the network to learn discriminative features at multiple scales while stabilizing gradient flow.

### 4. Experiments

Our $D^2$-Former model is evaluated on multiple medical image segmentation datasets, with implementation details and further results provided in subsequent sections.

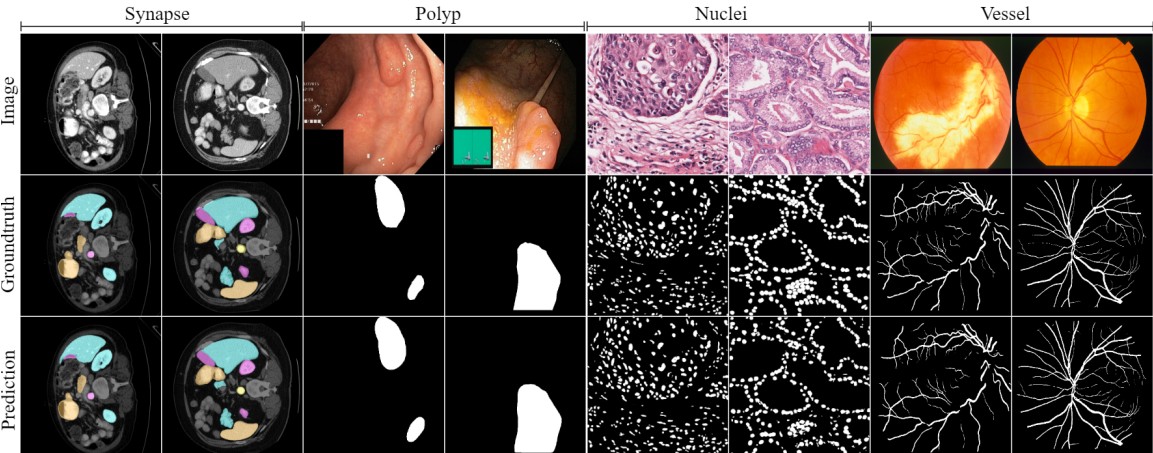

Figure 2: Qualitative results on synapse, polyp, nuclei, and retinal vessel datasets.

Table 1: Comparison to the SOTA methods on four polyp datasets.

| Methods | Kvasir-SEG | | CVC-ClinicDB | | CVC-300 | | CVC-ColonDB | |
|---|---|---|---|---|---|---|---|---|
| | Dice↑ | mIoU↑ | Dice↑ | mIoU↑ | Dice↑ | mIoU↑ | Dice↑ | mIoU↑ |
| U-Net(Ronneberger et al., 2015) | 81.8 | 74.6 | 82.3 | 75.5 | 71.0 | 62.7 | 51.2 | 44.4 |
| PraNet(Fan et al., 2020) | 89.8 | 84.0 | 89.9 | 84.9 | 87.1 | 79.7 | 70.9 | 64.0 |
| Polyp-PVT(Dong et al., 2023) | 91.7 | 86.4 | 93.7 | 88.9 | 90.0 | 83.3 | 80.8 | 72.7 |
| SegDINOv3 (Yang et al., 2025) | 87.6 | 80.6 | 94.0 | 88.2 | 90.2 | 85.0 | 81.2 | 73.2 |
| PAAN (Yi et al., 2024) | **94.2** | **89.7** | 93.4 | 88.4 | **92.6** | **86.9** | 78.6 | 71.6 |
| VM-UNet(J and S, 2024) | 91.3 | 85.6 | 92.6 | 87.1 | 88.6 | 81.8 | 79.8 | 71.2 |
| QueryNet (Chai et al., 2024) | 93.2 | 88.3 | 94.2 | **89.4** | 92.0 | 85.9 | 82.7 | 75.9 |
| SSformer (Wang et al., 2022) | 92.6 | 87.4 | 92.7 | 87.6 | 89.5 | 82.7 | 80.2 | 72.1 |
| ColnNet (Jain et al., 2023) | 92.6 | 87.2 | 93.0 | 88.7 | 90.9 | 86.3 | 79.7 | 72.9 |
| **D²-Former** | 93.4 | 87.3 | **94.2** | 88.9 | 92.0 | 86.1 | **82.8** | **76.3** |

## 4.1. Experimental Setup, Datasets, and Metrics

Our model was implemented in PyTorch and trained on an NVIDIA RTX 4090 GPU using 512×512 input images with a batch size of 8 for 100 epochs. We employed flipping augmentation and used the AdamW optimizer with an initial learning rate of 1e-4. Evaluation for polyp segmentation was performed on the Kvasir-SEG (Jha et al., 2020), CVC-ClinicDB (Bernal et al., 2015), and CVC-ColonDB (Tajbakhsh et al., 2015) datasets. We further conducted extensive experiments on several additional public benchmarks: the DRIVE (Staal et al., 2004) and STARE (Hoover et al., 2000) datasets for retinal vessel segmentation, the Synapse dataset (Chen et al., 2021b) for multi-organ abdominal CT segmentation (Synapse is handled 2D slice-wise with the standard split), and the MoNuSeg (Kumar et al., 2020) and CryoNuSeg (Mahbod et al., 2021) datasets for nuclei instance segmentation in histopathology. Performance was assessed using the Dice, mIoU, S-measure ($S_\alpha$), F-measure ($F_\beta^\omega$), E-measure ($E_p^{max}$), F1-Score, Acc, MAE, AJI, PQ, SP, and SE metrics. More details are in Appendix A.

Table 2: Quantitative results comparison on multi-organ abdominal CT dataset. Kidney Left (KL), Kidney Right (KR), Gallbladder (GB).

| Methods | Dice ↑ | mIoU ↑ | Aorta | GB | KL | KR | Liver | Pancreas | Spleen | Stomach |
|---|---|---|---|---|---|---|---|---|---|---|
| AttnUNet | 71.70 | 61.38 | 82.61 | 61.94 | 76.07 | 70.42 | 87.54 | 46.70 | 80.67 | 67.66 |
| SSFormer | 78.01 | 67.23 | 82.78 | 63.74 | 80.72 | 78.11 | 93.53 | 61.53 | 87.07 | 76.61 |
| PolypPVT | 78.08 | 67.43 | 82.34 | 66.14 | 81.21 | 73.78 | 94.37 | 59.34 | 88.05 | 79.40 |
| TransUNet | 77.61 | 67.32 | 86.56 | 60.43 | 80.54 | 78.53 | 94.33 | 58.47 | 87.06 | 75.00 |
| SwinUNet | 77.58 | 66.88 | 81.76 | 65.95 | 82.32 | 79.22 | 93.73 | 53.81 | 88.04 | 75.79 |
| PVT-CASCADE | 81.06 | 70.88 | 83.01 | 70.59 | 82.23 | 80.37 | 94.08 | 64.43 | 90.10 | 83.69 |
| TransCASCADE | 82.68 | 73.48 | 86.63 | 68.48 | 87.66 | 84.56 | 94.43 | 65.33 | 90.79 | 83.52 |
| PVT-EMCAD-B2 | 83.63 | 74.65 | 88.14 | 68.87 | 88.08 | 84.10 | 95.26 | 68.51 | 92.17 | 83.92 |
| **D²-Former** | **85.53** | **77.32** | **91.23** | **71.94** | **89.90** | **86.21** | **96.11** | **70.03** | **94.10** | **84.74** |

Table 3: Comparison to the SOTA methods on nuclei segmentation datasets.

| Methods | MoNuSeg | | | CryoNuSeg | | |
|---|---|---|---|---|---|---|
| | Dice ↑ | AJI ↑ | PQ ↑ | Dice ↑ | AJI ↑ | PQ ↑ |
| SAMUS (Lin et al., 2024) | 83.26 | 67.98 | 63.93 | 84.40 | 51.25 | 48.35 |
| SMMILe (Lu et al., 2023) | **89.20** | n/a | n/a | 87.60 | n/a | n/a |
| DenseU-Net (Kiran et al., 2022) | 83.21 | **78.61** | n/a | 83.50 | **79.42** | n/a |
| NucleiSeg (Swain et al., 2024) | 89.02 | n/a | n/a | n/a | n/a | n/a |
| NuSegDG (Lou et al., 2024) | 86.37 | 69.81 | 68.88 | 84.59 | 53.33 | 49.11 |
| **D²-Former** | 88.93 | 71.02 | **70.13** | **87.72** | 70.10 | **63.21** |

Table 4: Quantitative results on retinal vessel segmentation datasets.

| Methods | DRIVE | | | | STARE | | | |
|---|---|---|---|---|---|---|---|---|
| | F1 Score ↑ | Acc ↑ | SP↑ | SE ↑ | F1 Score ↑ | Acc ↑ | SP ↑ | SE↑ |
| TCDD-UNet (Nianzhu et al., 2024) | 82.65 | 96.98 | 98.12 | 84.21 | 81.63 | 97.40 | 98.37 | **82.24** |
| U-Net++ (Zhou et al., 2018) | 81.92 | 96.88 | 97.43 | 82.51 | 78.59 | 97.57 | 98.51 | 76.83 |
| DUNet (Qiangguo et al., 2019) | 82.37 | 95.66 | 97.92 | 79.70 | 81.43 | 96.41 | 98.81 | 73.70 |
| MambaUNet (Wang et al., 2024) | 81.92 | 95.51 | 97.73 | 80.48 | 78.96 | 96.11 | 98.75 | 73.19 |
| AttUKAN (Zeng et al., 2025) | 82.50 | 95.49 | 97.21 | 83.93 | 81.14 | 96.43 | 98.68 | 76.88 |
| **D²-Former** | **83.21** | **97.19** | **98.31** | **84.40** | **82.11** | **97.84** | **98.83** | 79.87 |

### 4.2. Experimental Results

**Polyp Segmentation:** The proposed D²-Former achieves competitive performance across four polyp segmentation benchmarks (Table 9). It obtains the highest Dice/IoU on CVC-ClinicDB (94.2/88.9) and CVC-ColonDB (82.8/76.3). On Kvasir-SEG, it ranks second (93.4/87.3) and remains highly competitive on CVC-300 (92.0/86.1). These results demonstrate D²-Former's strong generalization. A detailed analysis can be found in Appendix B.1.

**Nuclei Segmentation:** D²-Former delivers competitive nuclei segmentation performance, as summarized in Table 3. On CryoNuSeg, it achieves the highest Dice of 87.7, along with strong AJI (70.1) and PQ (63.2), outperforming existing methods across all reported metrics. For the MoNuSeg dataset, D²-Former attains a Dice of 88.9 with improved AJI (71.0) and PQ (70.1), remaining highly competitive with leading approaches such as SMMILe and

Table 5: Comprehensive reproducibility analysis across 5 independent training runs. Results are reported as mean ($\mu$) $\pm$ standard deviation ($\sigma$) for multiple evaluation metrics.

| Dataset | $\mu \pm \sigma$ | Dice ↑ | mIoU↑ | AJI↑ | PQ↑ | F1 Score↑ | Acc↑ | SP↑ | SE↑ |
|---|---|---|---|---|---|---|---|---|---|
| Kvasir | $\mu$ | 93.34 | 87.22 | n/a | n/a | n/a | n/a | n/a | n/a |
| | $\sigma$ | 0.12 | 0.15 | n/a | n/a | n/a | n/a | n/a | n/a |
| CVC-ClinicDB | $\mu$ | 94.08 | 88.81 | n/a | n/a | n/a | n/a | n/a | n/a |
| | $\sigma$ | 0.19 | 0.18 | n/a | n/a | n/a | n/a | n/a | n/a |
| Synapse | $\mu$ | 85.44 | 77.20 | n/a | n/a | n/a | n/a | n/a | n/a |
| | $\sigma$ | 0.20 | 0.23 | n/a | n/a | n/a | n/a | n/a | n/a |
| MoNuSeg | $\mu$ | 88.84 | n/a | 70.80 | 69.98 | n/a | n/a | n/a | n/a |
| | $\sigma$ | 0.12 | n/a | 0.31 | 0.18 | n/a | n/a | n/a | n/a |
| CryoNuSeg | $\mu$ | 87.59 | n/a | 70.02 | 63.11 | n/a | n/a | n/a | n/a |
| | $\sigma$ | 0.21 | n/a | 0.13 | 0.14 | n/a | n/a | n/a | n/a |
| DRIVE | $\mu$ | n/a | n/a | n/a | n/a | 83.10 | 97.14 | 98.19 | 84.28 |
| | $\sigma$ | n/a | n/a | n/a | n/a | 0.16 | 0.10 | 0.22 | 0.26 |
| STARE | $\mu$ | n/a | n/a | n/a | n/a | 82.01 | 97.75 | 98.75 | 79.80 |
| | $\sigma$ | n/a | n/a | n/a | n/a | 0.21 | 0.18 | 0.16 | 0.10 |

NucleiSeg. These results demonstrate that D²-Former is a robust and effective model for nuclei segmentation across diverse histopathological datasets.

**Retinal Vessel Segmentation:** D²-Former demonstrates strong performance on retinal vessel segmentation, enabling accurate delineation of fine vascular structures. As reported in Table 4, our model achieves the highest F1 scores on both DRIVE (83.21) and STARE (82.11), together with superior accuracy, specificity, and sensitivity compared to recent CNN-, Mamba-, and KAN-based baselines. These results highlight the robustness of D²-Former across retinal datasets, maintaining reliable performance on thin and low-contrast vessel structures.

**Multi-organ Abdominal CT Segmentation:** Our model exhibits exceptional generalization on the challenging multi-class Synapse dataset, a standard benchmark for abdominal CT segmentation with eight distinct organ classes. As shown in Table 2, our model achieves the highest mean Dice of 85.53 and mIoU of 77.32, significantly outperforming recent strong baselines such as TransCASCADE (Dice 82.68) (Rahman and Marculescu, 2023), PVT-CASCADE (Dice 81.06) (Titoriya and Singh, 2023), SwinUNet (Dice 77.58) (Cao et al., 2022a), and PVT-EMCAD-B2 (Dice 83.63) (Rahman et al., 2024). D²-Former demonstrates superior performance across all organs, particularly excelling in anatomically complex structures like the pancreas (70.03 Dice) and gallbladder (71.94 Dice), where other models often struggle. This indicates that the DINOv3-driven dual encoder, combined with adaptive routing via Softer-MoE, effectively captures both high-fidelity semantic features and fine-grained structural details, enabling robust segmentation across highly variable organ appearances and scales. Qualitative segmentation results across four distinct biomedical imaging domains—polyp, nuclei, retinal vessel, and multi-organ CT—are visualized in Figure 2, and additional predicted images from synapse and retinal vessel are shown in Appendix B.

**Statistical Validation and Reproducibility Analysis:** We evaluate reproducibility by repeating each experiment 5 times with different random seeds while keeping data splits and hyperparameters fixed. Table 5 reports the mean ($\mu$) and standard deviation ($\sigma$) of all metrics across runs. The low variance observed across datasets (0.12–0.21 for Dice and 0.15–0.23 for mIoU) indicates strong robustness to initialization. In addition, we performed

Table 6: Ablation study on dual-encoder components and fusion strategies. Encoder-Decoder (ED).

| Configuration | Kvasir-SEG | CVC-ClinicDB | CVC-ColonDB | DRIVE | STARE | Synapse |
|---|---|---|---|---|---|---|
| Baseline (Swin-ED) | 90.85 | 92.50 | 80.32 | 81.21 | 79.38 | 80.74 |
| + DINOv3 (concat) | 92.22 | 93.41 | 81.90 | 82.76 | 81.37 | 83.81 |
| + SF-GCA | 93.08 | 93.98 | 82.36 | 83.10 | 82.02 | 84.67 |
| + RAD (ours) | **93.42** | **94.21** | **82.83** | **83.21** | **82.11** | **85.53** |

Table 7: Ablation on Softer-MoE variants and stage placements.

| MoE Configuration | Kvasir | CVC-ClinicDB | Synapse | DRIVE |
|---|---|---|---|---|
| No MoE (baseline FFN) | 93.12 | 93.79 | 84.22 | 83.03 |
| MoE (Hwang et al., 2023) | 93.22 | 93.98 | 85.18 | 83.15 |
| Softer-MoE (stages 3–4) | 93.17 | 93.96 | 84.95 | 83.08 |
| Softer-MoE (stages 2–4) | 93.20 | 94.00 | 85.03 | 83.10 |
| Softer-MoE (4 experts) | 93.23 | 94.06 | 85.11 | 83.18 |
| Softer-MoE (8 experts) | 93.42 | 94.21 | 85.53 | 83.21 |

paired $t$-tests over the 5 matched-seed runs ($n = 5$) comparing $D^2$-Former to the Swin-ED baseline; improvements are significant across all reported metrics ($p < 0.001$).

## 4.3. Ablation Study

To comprehensively evaluate the contribution of each proposed component, we conduct systematic ablation experiments on the Kvasir-SEG, CVC-ClinicDB, CVC-ColonDB, DRIVE, STARE, and Synapse datasets. The baseline is a Swin-Transformer encoder-decoder with standard FFN blocks. All models are trained under identical settings (Sec. 4.1), and performance is measured by Dice score (%) and F1-score (for retinal vessel datasets).

### 4.3.1. Impact of Dual Encoder and Fusion Modules

We first investigate the contribution of the dual-encoder architecture and the proposed fusion mechanisms. The ablation study in Table 6 demonstrates the progressive improvement brought by each component of our dual-encoder design. The addition of the DINOv3 branch with simple concatenation enhances performance across all datasets, demonstrating dual-encoder superiority and confirming that the foundation model's high-fidelity dense features effectively complement the Swin Transformer's hierarchical local-global modeling. Replacing this concatenation with our Spatial-Frequency Gated Channel Attention (SF-GCA) module yields further gains, as it enables adaptive, channel-wise fusion of spatial details and frequency-domain context through learnable gating. Finally, integrating the Residual Attention Decoder (RAD) achieves the best results, validating that our decoder's multi-stage refinement with skip connections and attention mechanisms is essential for precise boundary delineation and robust generalization across diverse anatomical structures.

### 4.3.2. Role of Softer-MoE in Adaptive Feature Refinement

The ablation results in Table 7 confirm that replacing the static FFN with any MoE variant consistently improves performance across all datasets, validating the benefit of dy-

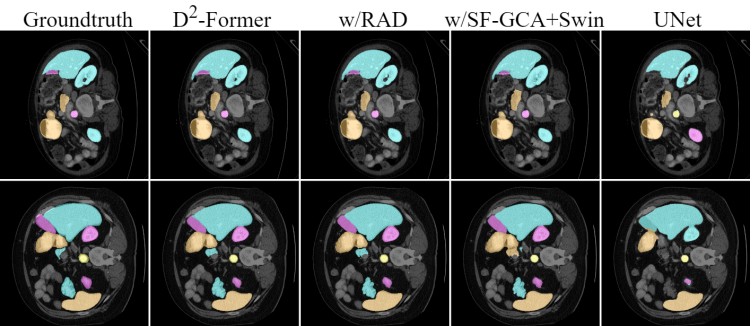

Figure 3: Visual ablation study on the synapse dataset.

Table 8: Ablation on decoder components and supervision.

| Decoder Variant | Kvasir | Synapse | CryoNuSeg |
|---|---|---|---|
| Plain U-Net decoder | 90.54 | 82.06 | 84.82 |
| + Skip connections only | 90.89 | 82.51 | 84.99 |
| + SCSE modules | 92.93 | 84.76 | 86.73 |
| + Residual Blocks (RB) | 93.13 | 85.41 | 87.42 |
| + Deep supervision (full RAD) | **93.42** | **85.53** | **87.71** |

namic expert routing for multi-scale anatomical structures. While base MoE and our initial Softer-MoE configurations (2 experts, stages 3-4 or 2-4) provide moderate gains, the optimal configuration employs 8 experts across stages 2–4, which yields the highest Dice scores (e.g., +0.30 on Kvasir and +1.31 on Synapse over baseline). This setup allows the model to apply soft, input-dependent expert blending from mid-to-high feature levels, enhancing adaptive refinement of both local boundaries and global context without over-specializing on low-level patterns.

### 4.3.3. Effect of Decoder Design and Supervision Strategy

We evaluate the contribution of the Residual Attention Decoder (RAD) and the deep supervision scheme. Table 8 presents the results. While the plain U-Net decoder effectively combines multi-scale features, it struggles to refine complex anatomical boundaries due to limited representational capacity. The introduction of SCSE modules addresses this by performing channel-spatial recalibration, yielding a marked improvement (e.g., +2.87 on Synapse) through better focus on salient structures. The Residual Block (RB) in the segmentation head further sharpens boundary delineation via local residual refinement. Finally, deep supervision stabilizes learning and improves performance, confirming RAD's effectiveness in segmenting complex anatomical regions (Figure 3).

### 4.3.4. Effect of DINO Backbone Variants

Table 9 reports a controlled comparison between DINOv2 and DINOv3 backbones within the proposed D²-Former framework. Across all benchmarks, DINOv3 variants consistently outperform their DINOv2 counterparts, demonstrating superior transferability of DINOv3's high-fidelity dense representations to medical image segmentation. In particular, DINOv3-

Table 9: Comparison to the different DINO variant models.

| Models | Kvasir-SEG | | Synapse | | DRIVE | | MoNuSeg | |
|---|---|---|---|---|---|---|---|---|
| | Dice↑ | mIoU↑ | Dice↑ | mIoU↑ | F1 Score↑ | Acc↑ | Dice↑ | AJI↑ |
| DINOv2-S | 92.04 | 86.26 | 85.02 | 76.75 | 82.13 | 96.31 | 87.56 | 69.23 |
| DINOv2-L | 92.87 | 87.03 | 85.32 | 77.11 | 82.97 | 97.08 | 88.52 | 70.50 |
| DINOv3-S+ | 93.42 | 87.31 | 85.53 | 77.32 | 83.21 | 97.19 | 88.93 | 71.02 |
| DINOv3-L | 93.81 | 88.79 | 85.92 | 78.98 | 84.11 | 97.63 | 89.21 | 71.15 |

Table 10: Overall model complexity.

| Model | Parameters (M)↓ | FLOPs (G)↓ | Inf-GPU (ms)↓ | Inf-CPU (ms)↓ | Model Size (MB)↓ |
|---|---|---|---|---|---|
| TransUNet | 86 | 11.7 | 22 | 280 | 328 |
| AttnUNet | 31.5 | 3.9 | 11 | 135 | 120 |
| $D^2$-Former | 119.9 | 66.21 | 84.8 | 265 | 457 |

S+ improves Dice and mIoU on Kvasir-SEG and Synapse, as well as F1-score and AJI on DRIVE and MoNuSeg, respectively. Although DINOv3-L achieves the best absolute performance, the improvement over DINOv3-S+ is marginal, while incurring significantly higher computational cost. Therefore, we adopt DINOv3-S+ for all experiments and ablation studies, as it provides the best balance between segmentation accuracy and efficiency.

### 4.3.5. Computational Efficiency Analysis

We compare the computational complexity of our model with representative baselines in Table 10. $D^2$-Former has 119.9M parameters and requires 66.21G FLOPs, making it larger and computationally heavier than lightweight models such as AttnUNet and TransUNet, and resulting in higher GPU and CPU inference latency. This overhead mainly stems from the dual-encoder architecture and the integration of Softer-MoE, which enhance representational capacity for complex anatomical structures. Despite the increased complexity, $D^2$-Former remains practically viable, achieving efficient GPU inference (84.8 ms per 256×256 image on an NVIDIA RTX 4090), and the added cost consistently translates into superior segmentation accuracy across diverse datasets.

## 5. Conclusion

Accurate medical image segmentation demands robust modeling of ambiguous boundaries and multi-scale anatomical variations. We present $D^2$-Former, a dual-encoder framework that integrates a Swin Transformer for hierarchical structural modeling with a DINOv3 foundation model for high-fidelity semantic features. To adaptively refine features, we introduce Softer-MoE blocks that softly route tokens to specialized experts, along with a Spatial-Frequency Gated Channel Attention (SF-GCA) module and a Residual Attention Decoder (RAD) for precise map reconstruction. Evaluated across nine public benchmarks covering polyp, retinal vessel, multi-organ CT, and nuclei segmentation, $D^2$-Former achieves state-of-the-art or highly competitive performance, demonstrating strong generalization across imaging modalities and anatomical scales. Our work establishes that Mixture-of-Experts guided dual Transformers offer a scalable and effective paradigm for clinical segmentation tasks. Future work will extend $D^2$-Former to 3D medical volumes, exploring volumetric Softer-MoE designs and spatio-temporal attention mechanisms for enhanced cross-modal understanding.

## Acknowledgments

This work was supported by the National Natural Science Foundation of China (Nos. 62401272 and 82441029).

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

## Appendix A. Experimental Detail

### A.1. Loss Function

The loss function is designed to address key challenges in medical image segmentation, including boundary ambiguity and background interference. We employ a weighted combination of three components:

$$\mathcal{L}_{\text{total}} = \alpha\mathcal{L}_{\text{BCE}} + \beta\mathcal{L}_{\text{Dice}} + \gamma\mathcal{L}_{\text{BL}} \tag{10}$$

where $\alpha = 0.5$, $\beta = 0.3$, $\gamma = 0.2$. The individual losses are defined as:

$$\mathcal{L}_{\text{BCE}} = -\frac{1}{N}\sum_{i=1}^{N}\left[y_i\log(p_i) + (1 - y_i)\log(1 - p_i)\right] \tag{11}$$

$$\mathcal{L}_{\text{Dice}} = 1 - \frac{2\sum_{i=1}^{N}y_ip_i}{\sum_{i=1}^{N}y_i + \sum_{i=1}^{N}p_i} \tag{12}$$

$$\mathcal{L}_{\text{BL}} = \frac{1}{N}\sum_{i=1}^{N}\mathbf{1}_{\{y_i=1\}} \cdot (1 - p_i)^2 \tag{13}$$

**Rationale for design:**

- *BCE Loss ($\mathcal{L}_{BCE}$)*: Handles class imbalance by assigning higher weights to minority classes (polyps), crucial for medical images where polyp regions are small.

- *Dice Loss ($\mathcal{L}_{Dice}$)*: Directly optimizes the overlap between prediction and ground truth, critical for boundary precision in polyp segmentation.

- *Boundary Loss ($\mathcal{L}_{BL}$)*: Focuses on boundary pixels (where $y_i = 1$) to improve edge delineation.

This combination effectively balances class imbalance, boundary accuracy, and overall segmentation quality.

## A.2. Experimental Setup

The experimental configuration is summarized in Table 11. The evaluation metrics include Dice, IoU, MAE, F$_{w\beta}$, S$_\alpha$, F1-score, Acc and E$_\phi$ as detailed in Section A.4. All experiments are conducted using the same training and testing protocols to ensure a fair comparison.

Table 11: Training and evaluation configuration.

| Parameter | Value |
|---|---|
| Input resolution | $512 \times 512$ |
| Batch size | 8 |
| Optimizer | AdamW ($\eta = 0.0001$, momentum=0.9) |
| Learning rate schedule | Step decay (0.1 at 50 epochs, 0.01 at 100 epochs) |
| Weight decay | $10^{-4}$ |
| Epochs | 100 |
| Data augmentation | Horizontal flip, rotation, scaling, Zoom in, Zoom out, Brightening |
| Hardware | NVIDIA RTX 4090 GPU |

## A.3. Datasets

Table 12: Dataset statistics (full URLs included).

| Dataset | # Images | Modality | Source |
|---|---|---|---|
| Kvasir-SEG | 1000 | Endoscopic | https://github.com/DebeshJha/Kvasir-SEG |
| CVC-ClinicDB | 612 | Endoscopic | https://www.kaggle.com/datasets/balraj98/cvcclinicdb |
| CVC-ColonDB | 380 | Endoscopic | https://www.kaggle.com/datasets/longvil/cvc-colondb |
| CVC-300 | 60 | Endoscopic | https://github.com/Polypproject/polyp_dataset |
| DRIVE | 40 | Color Fundus | https://github.com/hmoghimifam/DRIVE |
| STARE | 20 | Color Fundus | https://cecas.clemson.edu/~ahoover/stare/probing/index.html |
| Synapse | 30 (3779 slices) | Abdominal CT | https://github.com/Always70/Synapse |
| MoNuSeg | 30 (Train) | Histopathology | https://www.kaggle.com/datasets/tuanledinh/monuseg2018 |
| CryoNuSeg | 10 (per organ) | Histopathology | https://github.com/masih4/CryoNuSeg |

We evaluate D$^2$-Former on multiple medical image segmentation datasets as described below:

- **Kvasir-SEG**: Contains 1000 endoscopic images of polyps with pixel-wise annotations. Images exhibit varying polyp shapes, textures, and lighting conditions, making it challenging for boundary delineation.

- **CVC-ClinicDB**: A clinical dataset with 612 high-resolution colonoscopy images. Each image contains polyps with irregular shapes and blurred boundaries, along with normal tissues.

- **CVC-ColonDB**: Comprises 380 colonoscopy images with multiple polyps per image. Known for challenging visual conditions including poor contrast and complex backgrounds.

- **CVC-300**: Comprises 60 colonoscopy images with annotated polyp regions. This dataset is commonly used for polyp detection and segmentation tasks, featuring diverse visual conditions such as varying illumination, motion blur, and complex backgrounds.

- **DRIVE & STARE**: Widely used public benchmarks for retinal vessel segmentation. The DRIVE dataset contains 40 color fundus photographs with manual segmentation masks (Staal et al., 2004). The STARE dataset comprises 20 fundus images, also with vessel annotations, and is known for including pathological cases (Hoover et al., 2000). Both datasets are standard for evaluating algorithms' ability to segment thin and complex vascular networks.

- **Synapse**: A public multi-organ abdominal CT dataset from the MICCAI 2015 challenge. It contains 30 contrast-enhanced abdominal CT scans (comprising 3779 axial slices in total) with pixel-level annotations for 8 abdominal organs (e.g., aorta, gallbladder, pancreas). It serves as a standard benchmark for evaluating 3D medical image segmentation models, particularly on handling challenges such as inter-organ variability, ambiguous boundaries, and large slice-wise variations (Chen et al., 2021b).

- **MoNuSeg**: A public challenge dataset for multi-organ nuclei instance segmentation in digital pathology. The training set includes 30 H&E-stained tissue images from 7 organs, with annotations for over 21,000 nuclei. It serves as a key benchmark for evaluating the generalization of nuclei segmentation algorithms across different tissue types (Kumar et al., 2020).

- **CryoNuSeg**: The first fully annotated dataset for nuclei instance segmentation in cryosectioned H&E-stained histological images. It contains images from 10 human organs and is specifically designed to address the challenges posed by frozen section samples, which differ in appearance from the more common formalin-fixed paraffin-embedded (FFPE) samples (Mahbod et al., 2021).

**Data Processing:** All datasets are preprocessed to $512 \times 512$ resolution. For polyp segmentation, we use 1450 images (550 from CVC-ClinicDB and 900 from Kvasir) for training and 62 from CVC-ClinicDB, 100 from Kvasir, 62 from CVC-300 and 380 images from CVC-ColonDB for testing. Standard data augmentation (horizontal flip, rotation, and scaling) is applied to all training images to enhance model generalization. For the additional segmentation tasks, we adhered to the standard training and testing splits defined by the respective benchmark datasets to ensure a fair comparison. Specifically, for DRIVE and STARE, we used the standard split (20 train/20 test) and the common practice of training

on DRIVE and testing on STARE for cross-dataset evaluation. For Synapse, MoNuSeg, and CryoNuSeg, we followed the official challenge protocols and data splits.

## A.4. Evaluation Metrics

We employ standard metrics to comprehensively assess segmentation performance:

- **Dice (Mean Dice Coefficient)**:

$$\text{Dice} = \frac{2|X \cap Y|}{|X| + |Y|}$$

  Measures the overlap between predicted ($X$) and ground truth ($Y$) masks. Higher values indicate better segmentation accuracy, particularly for boundary precision.

- **mIoU (Mean Intersection over Union)**:

$$\text{mIoU} = \frac{|X \cap Y|}{|X \cup Y|}$$

  Evaluates the ratio of intersection to union of predicted and ground truth masks. A standard metric for segmentation tasks, with higher values indicating better overlap.

- **F$_\beta$ (F-measure)**:

$$\text{F}_\beta = \frac{(1 + \beta^2) \cdot \text{Precision} \cdot \text{Recall}}{\beta^2 \cdot \text{Precision} + \text{Recall}} \quad (\beta^2 = 0.5)$$

  Balances precision and recall with a bias toward recall ($\beta^2 < 1$) to emphasize detection of small structures. Higher values indicate better trade-off between false positives and false negatives.

- **S$_\alpha$ (Structure-measure)**:

$$S_\alpha = \alpha \cdot SO + (1 - \alpha) \cdot SR \quad (\alpha = 0.06)$$

  Combines object similarity ($SO$) and region similarity ($SR$) to evaluate structural consistency. Higher values indicate better preservation of anatomical structures.

- **E$_{max}$ (Enhanced-measure)**:

$$E_{max} = \frac{(1 + \phi^2) \cdot SO \cdot SR}{\phi^2 \cdot SO + SR} \quad (\phi^2 = 15)$$

  Refines the S$_\alpha$ metric by emphasizing region coverage. Higher values indicate more complete segmentation of target regions.

- **MAE (Mean Absolute Error)**:

$$\text{MAE} = \frac{1}{N} \sum_{i=1}^{N} |p_i - y_i|$$

  Quantifies the average absolute difference between predicted ($p_i$) and ground truth ($y_i$) pixels. Lower values indicate higher pixel-level accuracy, particularly important for boundary delineation.

- **F1-Score (F-Measure)**:

$$F_1 = \frac{2 \cdot \text{Precision} \cdot \text{Recall}}{\text{Precision} + \text{Recall}}$$

Balances precision and recall, providing a harmonic mean that is especially useful when class distribution is imbalanced, as often seen in vessel segmentation tasks.

- **Accuracy (Acc)**:

$$\text{Accuracy} = \frac{\text{TP} + \text{TN}}{\text{TP} + \text{TN} + \text{FP} + \text{FN}}$$

Measures the overall proportion of correctly classified pixels (both vessel and background), offering a global assessment of segmentation correctness.

- **Aggregated Jaccard Index (AJI):**

$$AJI = \frac{\sum_i |G_i \cap P_{m(i)}|}{\sum_i |G_i \cup P_{m(i)}| + \sum_{k \in U} |P_k|}$$

Measures instance-level segmentation quality by penalizing over- and under-segmentation, where $G_i$ denotes a ground-truth instance, $P_{m(i)}$ its matched prediction, and $U$ the set of unmatched predicted instances. Higher values indicate better instance delineation.

- **Panoptic Quality (PQ):**

$$PQ = \frac{\sum_{(p,g) \in TP} \text{IoU}(p, g)}{|TP| + \frac{1}{2}|FP| + \frac{1}{2}|FN|}$$

Evaluates both segmentation quality and detection accuracy for instance segmentation, combining recognition and mask quality. Higher values indicate better overall instance segmentation performance.

- **Specificity (SP):**

$$SP = \frac{TN}{TN + FP}$$

Measures the proportion of correctly identified background pixels, reflecting the model's ability to suppress false positives. Higher values indicate better background discrimination.

- **Sensitivity (SE):**

$$SE = \frac{TP}{TP + FN}$$

Measures the proportion of correctly detected foreground pixels, reflecting the model's ability to capture target structures. Higher values indicate better foreground recall.

These metrics collectively assess accuracy, boundary precision, structural similarity, and robustness to class imbalance, providing a comprehensive evaluation of segmentation performance across diverse medical imaging tasks.

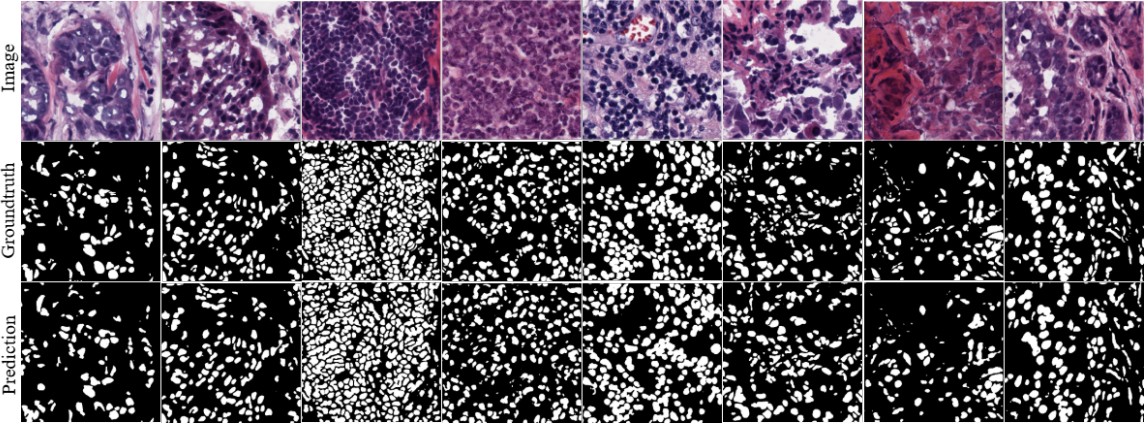

Figure 4: Additional qualitative results on nuclei image segmentation.

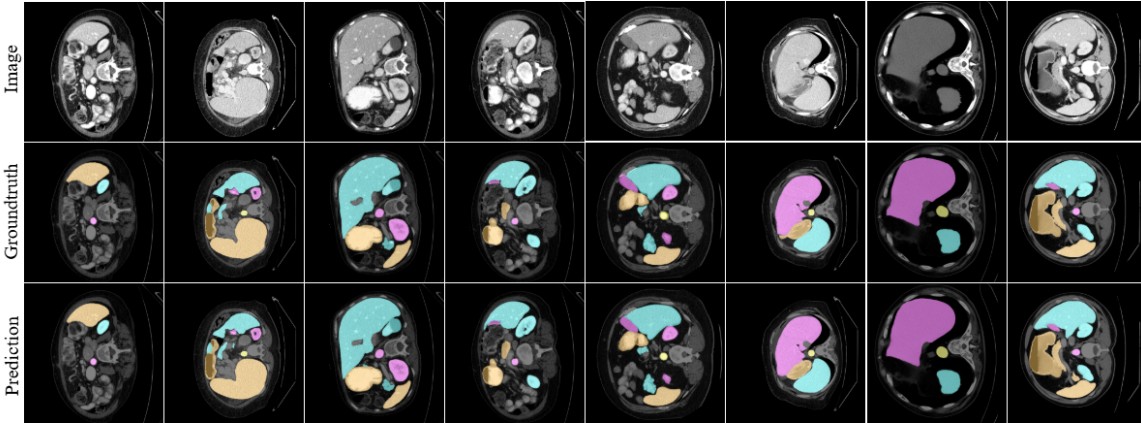

Figure 5: Additional qualitative results on Synapse image segmentation.

## Appendix B. Additional Results

### B.1. Appendix: Detailed Polyp Segmentation Results

Comprehensive quantitative results for polyp segmentation across four standard benchmarks are presented in Tables 11 and 12. Beyond the Dice and mIoU scores reported in the main text, our D$^2$-Former framework achieves superior or highly competitive performance on multiple additional metrics, including the weighted F-measure ($F_{sp}$), structure-measure ($S_\alpha$), enhanced-alignment measure ($E_p^{max}$), and mean absolute error (MAE).

Specifically, on the seen-domain datasets, Kvasir-SEG and CVC-ClinicDB, D$^2$-Former attains the highest $F_{sp}$ scores of 94.2 and 94.3, respectively, and competitive scores on $S_\alpha$ and $E_p^{max}$. On the more challenging unseen-domain datasets, CVC-300 and CVC-ColonDB, our model demonstrates strong generalization, leading in $F_{sp}$ (93.8 on CVC-300, 85.6 on CVC-ColonDB) while achieving the best $S_\alpha$ score (95.5) on CVC-300. The results across

Table 13: Quantitative results comparison of D$^2$-Former with SOTA methods on Kvasir-SEG and CVC-ClinicDB datasets.

| Methods | Kvasir-SEG | | | | | | CVC-ClinicDB | | | | | |
|---|---|---|---|---|---|---|---|---|---|---|---|---|
| | mDice ↑ | mIoU ↑ | $F_{sp}$ ↑ | $S_\alpha$ ↑ | $E_p^{max}$ ↑ | MAE ↓ | mDice ↑ | mIoU ↑ | $F_{sp}$ ↑ | $S_\alpha$ ↑ | $E_p^{max}$ ↑ | MAE ↓ |
| U-Net | 81.8 | 74.6 | 79.4 | 85.8 | 89.3 | 5.5 | 82.3 | 75.5 | 81.1 | 88.9 | 95.4 | 1.9 |
| PraNet | 89.8 | 84.0 | 88.5 | 91.5 | 94.8 | 3.0 | 89.9 | 84.9 | 89.6 | 93.6 | 97.9 | 0.9 |
| Polyp-PVT | 91.7 | 86.4 | 91.1 | 92.5 | 95.6 | 2.3 | 93.7 | 88.9 | 93.6 | 94.9 | 98.5 | 0.6 |
| VM-UNet | 91.3 | 85.6 | 90.2 | 91.8 | 95.8 | 2.7 | 92.6 | 87.1 | 92.7 | 93.3 | 97.1 | 0.9 |
| ColnNet | 92.6 | 87.2 | 93.9 | 92.6 | 97.9 | 2.0 | 93.0 | 88.7 | 94.0 | 95.2 | 98.7 | 0.6 |
| D$^2$-Former | 93.4 | 87.3 | 94.2 | 93.6 | 96.1 | 2.2 | 94.2 | 88.9 | 94.3 | 95.5 | 98.2 | 0.6 |

Table 14: Quantitative results comparison of D$^2$-Former with SOTA methods on Unseen CVC-300 and CVC-ColonDB datasets.

| Methods | CVC-300 | | | | | | CVC-ColonDB | | | | | |
|---|---|---|---|---|---|---|---|---|---|---|---|---|
| | mDice ↑ | mIoU ↑ | $F_{sp}$ ↑ | $S_\alpha$ ↑ | $E_p^{max}$ ↑ | MAE ↓ | mDice ↑ | mIoU ↑ | $F_{sp}$ ↑ | $S_\alpha$ ↑ | $E_p^{max}$ ↑ | MAE ↓ |
| U-Net | 71.0 | 62.7 | 68.4 | 84.3 | 87.6 | 2.2 | 51.2 | 44.4 | 49.8 | 71.2 | 77.6 | 6.1 |
| PraNet | 87.1 | 79.7 | 84.3 | 92.5 | 97.2 | 1.0 | 70.9 | 64.0 | 69.6 | 81.9 | 86.9 | 4.5 |
| Polyp-PVT | 90.0 | 83.3 | 88.4 | 93.5 | 97.3 | 0.7 | 80.8 | 72.7 | 79.5 | 86.5 | 91.3 | 3.1 |
| VM-UNet | 88.6 | 81.8 | 84.9 | 92.1 | 96.8 | 0.9 | 79.8 | 71.2 | 78.2 | 86.1 | 90.4 | 3.6 |
| CoinNet | 90.9 | 86.3 | 88.1 | 94.2 | 98.9 | 0.5 | 79.7 | 72.9 | 78.9 | 87.5 | 89.7 | 2.2 |
| D$^2$-Former | 92.0 | 86.1 | 93.8 | 95.5 | 98.1 | 0.6 | 82.8 | 76.3 | 85.6 | 89.8 | 93.3 | 1.9 |

this broader set of evaluation metrics consistently reinforce the robustness and precision of the proposed model for polyp segmentation across diverse clinical scenarios.

## B.2. Appendix: Additional results from Synapse and nuclei datasets

Figure 5 and Figure 4 presents qualitative segmentation results on the Synapse multi-organ CT and nuclei datasets. The visualizations demonstrate that D$^2$-Former accurately delineates complex organ boundaries in abdominal CT scans, including challenging structures like the pancreas and gallbladder, and effectively segments individual nuclei in dense histopathology images. These results highlight the model's strong generalization across both macro-scale anatomical and micro-scale cellular segmentation tasks.

## Appendix C. Responsible AI and Deployment Considerations

Although we evaluate on multiple public benchmarks, these datasets may not fully represent the diversity of real clinical settings (e.g., scanner protocols, institutions, patient populations, and rare pathologies), and annotations can contain inter-observer variability. As a result, performance may degrade under domain shift and may exhibit systematic errors for under-represented cases. In deployment, segmentation errors (e.g., over-segmentation that inflates lesion extent or false positives on normal tissue) could contribute to unnecessary follow-up or intervention if used without clinical oversight. We therefore emphasize that D$^2$-Former is intended as decision support; prospective validation, site-specific calibration/thresholding, and human-in-the-loop review are necessary before any clinical use.

