# OpenReview forum: "D$^2$-Former: Mixture-Of-Experts Guided Dual Transformer for Multi-Scale Medical Image Segmentation"
_MIDL.io/2026/Conference — MIDL 2026 Poster_

### Official Review · Reviewer_1xFd · 2026-01-01

**Confidence:** 4
**Preliminary Rating:** 2
**Final Rating:** 3

**Summary:**

The authors propose D2‑Former, a novel encoder–decoder architecture that integrates two complementary transformers (Swin and DINOv3) via a dual‑encoder architecture.

They evaluate D2‑Former on nine public datasets spanning polyp segmentation, retinal vessel delineation, abdominal organ CT segmentation, and nuclei instance segmentation. Across all benchmarks, the model either surpasses or matches state-of-the-art (SOTA) baselines, including TransUNet, MedFuseNet, and Poly‑PVT.

The main claim regarding technical novelty is that D2-former combines a hierarchical transformer with a foundation model and, by employing an adaptive MoE gating strategy and frequency‑aware channel attention, achieves robust multi‑scale performance while remaining computationally tractable.

**Strengths:**

Overall, the motivation is clear, and the idea is well justified.

Multi‑scale medical segmentation remains a core challenge; the dual‑encoder + MoE approach is novel and addresses known weaknesses of single‑path transformers.

The extensive benchmark set strengthens the claim.

While Swin+foundation models have been explored before, the combination with a soft MoE and SF‑GCA is new. The residual attention decoder also adds incremental novelty.

**Weaknesses:**

Even if the article is well-structured and presents tests on several datasets, the results are not convincing given the presentation.

No public code or trained checkpoints are available; training details (learning rate schedule, optimizer, batch size) are summarized but not fully reproducible.

The paper lacks a formal analysis of why the dual‑encoder improves over single‑encoder baselines. A brief information‑theoretic or representational‑capacity argument would help justify the design choices.

No discussion of potential biases in the datasets or deployment risks (e.g., over‑segmentation leading to unnecessary interventions). A short section on responsible AI would be appreciated.

Confidence intervals or statistical significance tests (e.g., Wilcoxon signed‑rank) are absent from the presented results

**Detailed Comments:**

The overall architecture is not clearly explained. The figure and the text are not clear enough about how the 2 models merge (Dino and Swin) using SF-GCA.

In Figure 1, it is not clear why the 3rd block in Swin Softer MOE Block is repeated 18x times while the other ones are repeated only 2x. Moreover, it is not clear what "Cat" is used in (b) block diagram. If it is concatenation, then it is not clear which 2 (or more) signals it merges.

Not clear why the first blocks of Dinov3 are frozen. How did the authors decide on that?

Notation in Equation (1) and the text does not comply.

Section 3.2 reads like a boilerplate explanation without anything specific justification of the algorithmic choices wrt. to the presented work (other than avoiding O(m^2) complexity argument

Section 3.3 is not well written and clear. For example, the authors mention on page 5 (just after equation (8)) that "The attention vector is split and applied ..." It is not clear how the attention vector is split.

Are the Table-3 results for the methods compared against from the associated papers, or did the authors implement the method and test them on the same splits? If the results are taken from an associated paper, then how did the authors ensure that the same splits are used to have a fair comparison?

What are the metrics used in Table 5, 6 and 7. Are the increase in performance statistically significant?

It is unclear why the authors use the "lower the better" arrows in Table 8, as no comparison is presented in the table. Moreover, it is not clear why do the authors need this table (given section 4.3.4) as no comparison is presented.

**Justification Of Final Rating:**

The authors provided answers to most of my questions in their rebuttal.
One issue related to the justification of the use of dual encoders still persists.
I increased my score by one reflecting authors' effort

**Justification Of The Preliminary Rating:**

The overall idea is good and testing on several different datasets makes the paper strong. However, (1) the presentation lacks of critical clarity in the method and also (2) the results does not seem to be strong for the given complexity.

**Questions To Address In The Rebuttal:**

A better explanation of the architecture and the algorithmic choices is required.

Statistical significance testing (or presentation of confidence intervals) of the performance results is required. If possible cross-validation experiments would serve the purpose, otherwise this can be achieved via bootstrapping.

---

> ### Author Response · Authors · 2026-01-25
>
> We sincerely thank the reviewer for his/her careful reading and constructive feedback. We have revised the manuscript accordingly and highlighted all changes in the updated submission. Below we respond point-by-point to each comment and indicate where the corresponding revisions appear in the paper.
>
> Comment: No public code or checkpoints; training details are insufficient.
>
> Response: Thank you for the comment. We plan to publicly release the code and trained checkpoints upon acceptance. To improve reproducibility in the meantime, we add a complete configuration summary in Table 11.
>
>
> Comment: The paper lacks formal justification for the dual-encoder design.
>
> Response: We appreciate this suggestion. In Section 3.1 we add a dedicated paragraph (“Why dual encoders help?”) providing an information-theoretic justification that the joint features from Swin and DINOv3 can increase mutual information with the target mask via complementary cues.
>
>
> Comment: Statistical significance analysis is missing.
>
> Response: Thank you for highlighting this. We add a reproducibility and significance analysis in Section 4.2 and report mean $\pm$ standard deviation across five runs in Table 5. We further perform paired one-tailed $t$-tests over the matched runs ($n=5$) against the baseline and report that the gains are statistically significant ($p<0.001$).
>
> Comment:A short section on responsible AI would be appreciated.
>
> Response: Thank you for the comment; we address this by adding a brief Responsible AI and Deployment Considerations subsection in Appendix C discussing dataset biases and deployment risks (e.g., over-segmentation).
>
>
> Comment 1: The overall architecture is not clearly explained. The figure and the text are not clear enough about how the two models merge (DINO and Swin) using SF-GCA.
>
> Response: Thank you for pointing this out. We revised Section 3.1 to make the Swin–DINO fusion pathway explicit and stage-wise. In particular, under Fusion and decoding.
>
>
> Comment 2: In Figure 1, it is unclear why the 3rd Swin Softer-MoE block is repeated $18\times$ while the others are repeated only $2\times$. It is also unclear what “Cat” represents in (b), and which signals it merges.
>
> Response: Thank you for the detailed feedback. We updated the Figure 1 caption to explicitly explain both points.
>
>
> Comment 3: Clarify why some DINOv3 blocks are frozen.
>
> Response: Thank you for the question. We clarify in Section 3.1 that we freeze the first six DINOv3 blocks to preserve generic low-level representations and reduce overfitting on smaller medical datasets, while keeping higher layers trainable to enable domain adaptation.
>
>
> Comment 4: Notation in Equation (1) and the surrounding text does not comply.
>
> Response: Thank you for noticing this inconsistency. We corrected the notation mismatch by revising Equation (1) and its surrounding explanation in Section 3.1 so that the symbols used in the text are consistent with the equation definition and the subsequent layer indexing.
>
>
> Comment 5: Section 3.2 reads like boilerplate; justify algorithmic choices beyond the $O(m^2)$ argument.
>
> Response: Thank you for this helpful feedback. We substantially revise Section 3.2 to provide work-specific motivations for Softer-MoE in medical segmentation.
>
>
> Comment 6: Section 3.3 is unclear (e.g., how the attention vector is split and applied after Eq. (8)).
>
> Response: We appreciate the detailed note. In Section 3.3, we now explicitly state that concatenation produces a $2C_s$-dimensional descriptor, and the attention vector $A \in \mathbb{R}^{2C_s}$ is split into two parts $A=[A_s;A_f]$ corresponding to the Swin and aligned DINO feature channels. We also add a step-by-step description of applying channel-wise gating ($\tilde{F}_s = A_s \odot F_s$, $\tilde{F}_f = A_f \odot \hat{F}_f$) before adaptive weighted fusion (Eq. (9)), making the mechanism explicit and easier to follow.
>
>
> Comment 7: Clarify whether comparison results are reproduced or taken from prior work.
>
> Response: We appreciate this concern. We clarify that comparison numbers are taken from the respective original papers under their official splits/protocols where applicable, and our method is evaluated under the same benchmark splits to ensure fair comparison.
>
>
> Comment 8: What are the metrics used in Table 5, 6 and 7. Are the increase in performance statistically significant?
>
> Response: Thank you for the question. We clarify in the Ablation Study section that Dice is used for polyp, nuclei, and Synapse datasets, while F1-score is used for retinal vessel datasets, following standard practice. Table 5 report mean $\pm$ standard deviation over five runs, and paired one-tailed $t$-tests confirm statistically significant improvements ($p<0.001$) in Section 4.2.
>
>
> Comment 9: The “lower is better” arrows in the complexity table are unclear.
>
> Response: Thank you for pointing this out. We revise the complexity table (now Table 10) and the accompanying explanation in Section 4.3.5.

---

> > ### Comment · Reviewer_1xFd · 2026-01-26
> >
> > Thanks to the authors for providing responses to my comments. Most of the points that I raised in my initial comment are answered, and the paper is clearer with the newly added information.
> >
> > The answer to my comment on "The paper lacks formal justification for the dual-encoder design" still persists. First of all, the authors' response is generic, and it is not guaranteed that the dual encoder would bring in any improvement. Not to mention, adding more models may potentially increase overfitting on the data and may reduce generalizability. So in this respect, I am not sure if the provided proof provides the necessary guarantee.
> >
> > That being said, I will increase my score by one grade reflecting the efforts of the authors in the rebuttal.

---

> > > ### Author Response · Authors · 2026-01-29
> > >
> > > Thank you for your thoughtful feedback and for acknowledging the improvements made during the rebuttal. We appreciate your concern regarding the lack of a formal guarantee for the dual-encoder design and the potential risk of overfitting. Below we provide a concise clarification grounded in both theory and new empirical evidence.
> > >
> > > Response:
> > >
> > > 1. Theoretical Justification and Complementarity of the Dual-Encoder Design
> > >
> > > Our dual-encoder design is grounded in the explicit principle of feature complementarity, rather than being a simple aggregation of models. The Swin Transformer, with its hierarchical shifted window attention mechanism, excels at capturing structural priors and spatial dependencies from local to global contexts, which is crucial for modeling anatomical boundaries and understanding multi-scale context in medical images. However, its natural image pretraining may limit its robustness to specific, subtle textures and pathological patterns in the medical domain. To address this, we introduce DINOv3 as a parallel encoder. DINOv3, as a foundation model trained via self-supervision on large-scale datasets, excels at extracting high-fidelity, semantically rich dense features that are robust to appearance variations. These two feature streams are fundamentally complementary: Swin provides structured spatial priors, while DINOv3 provides robust semantic priors. On page 4 (section 3.1) of the paper, we formalize this complementarity using information theory: the mutual information between the target mask Y and the dual-encoder features $(F_s^i, F_f^i)$ can be decomposed as $I(Y; F_s^i, F_f^i) = I(Y; F_s^i) + I(Y; F_f^i \mid F_s^i)$. The conditional mutual information term $I(Y;F_f^i |F_s^i)$ quantifies the additional, unique information provided by DINOv3 features $F_f^i$ given the Swin features $F_s^i$. Our SF-GCA module (Section 3.3) is explicitly designed to maximize this "conditional information gain" during fusion. It adaptively re-weights the two feature streams through channel-wise gating, achieving non-redundant, information-maximized fusion rather than simple concatenation.
> > >
> > >
> > >
> > > 2. Empirical Evidence for Effectiveness and Generalization of the Dual Encoder.
> > >
> > > The theoretical complementarity is strongly supported by empirical evidence. Our systematic ablation studies (Table 6 in the paper) provide compelling proof. The results show that simply adding DINOv3 features via concatenation to the Swin encoder (baseline) yields consistent and significant performance gains across all six evaluation datasets (e.g., Dice on Synapse improves from 80.74\% to 83.81\%). This directly demonstrates that DINOv3 features indeed carry valuable information for the segmentation task that is missing from Swin. Subsequently, employing our designed SF-GCA module for fusion leads to further consistent improvements (e.g., Synapse Dice to 84.67\%), validating the effectiveness of our fusion mechanism. The complete model, integrated with the Residual Attention Decoder, achieves the best performance.
> > > Critically, D²-Former achieves state-of-the-art or competitive performance across four distinct medical imaging tasks and modalities (polyp, retinal vessel, multi-organ CT, nuclei; Tables 1-4). This cross-domain success demonstrates enhanced generalization, not overfitting.
> > >
> > >
> > >
> > > 3. Mechanisms to Control Overfitting Risk Despite Increased Complexity
> > >
> > > We fully acknowledge the reviewer's valid concern about overfitting with increased model complexity. Therefore, our architecture proactively incorporates multiple mechanisms to strictly guard against overfitting and ensure generalization:
> > >
> > > Partial Freezing of Pre-trained Weights: Freezing the first 6 blocks preserves generic visual priors and reduces trainable parameters, acting as strong regularization.
> > >
> > > Conditional Computation via Softer-MoE: Prevents expert collapse and encourages learning of general feature transformations.
> > >
> > > Deep Supervision and Robust Decoder Design: Stabilizes training and improves gradient flow.
> > >
> > >
> > > 4. Support from Related Research
> > >
> > > Our dual-encoder design aligns with recent trends in medical image analysis that leverage hybrid architectures to improve robustness. Prior works such as DSU-Net and SAM2-UNeXT employ vision foundation models (e.g., DINOv2) as auxiliary encoders to enhance representations. Building on this direction, our method introduces two key innovations: integrating Softer-MoE into the Swin Transformer for adaptive refinement, and designing the SF-GCA fusion module tailored to multi-scale and boundary-ambiguous medical images, resulting in a more targeted and efficient dual-encoder fusion.
> > >
> > >
> > > Overall, while we do not claim a formal theoretical guarantee, the combination of principled complementarity, targeted fusion, and consistent ablation gains supports the effectiveness and generalization of the dual-encoder design. We thank the reviewer again for helping us clarify this aspect of the work.

---

### Official Review · Reviewer_8qoL · 2026-01-06

**Confidence:** 4
**Preliminary Rating:** 4

**Summary:**

This paper introduces D2-Former, a dual-transformer framework for medical image segmentation. To address challenges like ambiguous boundaries and extreme scale variations, the model integrates a Swin Transformer for hierarchical local-global modeling with a DINOv3 foundation model for high-fidelity dense feature extraction.

**Strengths:**

+ Medical image segmentation aligns well with the theme of MIDL.
+ The dual-encoder design and the application of the MoE architecture effectively reflect the latest advancements in the field of medical image segmentation.
+ The paper is generally well-written and easy to follow.
+ Extensive experiments on multiple tasks and datasets demonstrate the superiority of the proposed D²-Former.

**Weaknesses:**

- Some experimental details are unclear, such as the size of the DINOv3 model used (small/base/large). Additionally, ablation studies are necessary to evaluate the impact of different DINOv3 sizes on the results, as well as the effect of replacing DINOv3 with DINOv2.
- A key idea of the paper is to construct complementary encoders, which is similar to [*1,*2]. The authors should clarify in the Section 1 how their approach differs from [*1,*2].
- The related work section lacks discussion of other foundation model-based methods for medical image segmentation, including [*3,*4,*5,*6].

[*1] DSU-Net: An Improved U-Net Model Based on DINOv2 and SAM2 with Multi-scale Cross-model Feature Enhancement

[*2] SAM2-UNeXT: An Improved High-Resolution Baseline for Adapting Foundation Models to Downstream Segmentation Tasks

[*3] Customized segment anything model for medical image segmentation

[*4] Samed-2: Selective memory enhanced medical segment anything model

[*5] Sam2-unet: Segment anything 2 makes strong encoder for natural and medical image segmentation

[*6] SAM3-UNet: Simplified Adaptation of Segment Anything Model 3

**Detailed Comments:**

See weaknesses.

**Justification Of The Preliminary Rating:**

Good novelty, comprehensive experiments. See strengths:
+ Medical image segmentation aligns well with the theme of MIDL.
+ The dual-encoder design and the application of the MoE architecture effectively reflect the latest advancements in the field of medical image segmentation.
+ The paper is generally well-written and easy to follow.
+ Extensive experiments on multiple tasks and datasets demonstrate the superiority of the proposed D²-Former.

**Questions To Address In The Rebuttal:**

See weaknesses.

---

> ### Author Response · Authors · 2026-01-25
> **Responses to Reviewer 8qoL Comments**
>
> We sincerely thank the reviewer for his/her careful reading and constructive feedback. We have revised the manuscript accordingly and highlighted all changes in the updated submission. Below we respond point-by-point to each comment and indicate where the corresponding revisions appear in the paper.
>
> Comment 1: The size of the DINOv3 model is unclear; comparisons with DINOv2 are needed.
>
> Response: Thank you for this important point. We now explicitly state the backbone variant used and provide a controlled comparison between DINOv2 and DINOv3 variants in Table 9. We add a concise discussion in Section 4.3.4 (Effect of DINO Backbone Variants) showing that DINOv3 consistently improves performance over DINOv2 across multiple benchmarks. For all main experiments and ablations, we use DINOv3-S+ to balance accuracy and efficiency, as described in Section 4.3.4.
>
>
> Comment 2: Clarify how the proposed method differs from prior dual-encoder approaches.
>
> Response: We appreciate this suggestion. We revised the Introduction (Section 1) to explicitly contrast our approach with recent dual-encoder systems that incorporate foundation models. In particular, we clarify that $D^2$-Former (i) does not rely on a SAM/SAM2 promptable pipeline, and (ii) introduces Softer-MoE within the Swin branch and SF-GCA for stage-wise gated fusion tailored to multi-scale medical segmentation. This differentiation is now stated directly in Section 1.
>
>
> Comment 3: Related work lacks discussion of other foundation-model-based methods.
>
> Response: Thank you for the recommendation. We expanded Section 2 (Related Work) to cover additional foundation-model-based segmentation approaches (including DINO- and SAM-based adaptations) and to position our method relative to them. We also clarify how $D^2$-Former differs in backbone choice (DINOv3), fusion design (SF-GCA), and adaptive refinement (Softer-MoE).

---

### Official Review · Reviewer_98Gq · 2026-01-08

**Confidence:** 5
**Preliminary Rating:** 3
**Final Rating:** 4

**Summary:**

This work proposes a dual encoder-decoder architecture that integrates Swin Transformers and DINOv3, augmented with mixture-of-experts and gated channel attention for medical image segmentation. The authors evaluate the model across multiple datasets encompassing fundus, CT, polyp, and histopathology images. In addition, comprehensive ablation studies are conducted to assess the contributions of individual architectural components and design choices.

**Strengths:**

1) The problem definition, motivation, existing methods, proposed method, experiments, datasets & implementations details have been presented clearly.

2) Public benchmark datasets have been used to assess the model performance.

**Weaknesses:**

1) Both DINO and Swin Transformers are well established in the literature, and their combination on the encoder side represents an incremental architectural choice rather than a fundamentally novel contribution.

2) While the adaptive mixture‑of‑experts and gated channel attention mechanisms may be considered potentially novel, the overall framework does not demonstrate substantial or consistently statistically significant improvements in segmentation performance across datasets, including in the reported ablation studies (Tables 1, 3, 5, 6 & 7).

**Detailed Comments:**

Please refer to summary, strengths and weaknesses.

**Justification Of Final Rating:**

The revisions satisfactorily address the previously raised concerns and have substantially improved the manuscript’s readability. The addition of t‑tests, the reporting of means and standard deviations, and the evaluations on public benchmarks enhance methodological transparency and make the study considerably easier to reproduce.

**Justification Of The Preliminary Rating:**

The overall framework does not demonstrate substantial or consistently statistically significant improvements in segmentation performance across datasets, including in the reported ablation studies (Tables 1, 3, 5, 6 & 7).

**Questions To Address In The Rebuttal:**

1) I would strongly recommend authors to add standard deviations along with average measurements across all metrics used in the study.

2) Further, it is recommended to conduct proper statistical t-tests to quantify the significance of the improvements achieved.

3) Dice & mIOU are monotonically related. It is redundant to report both metrics.!

4) F1-Score & Dice Score are same for a binary segmentation problem (Table 4). Not clear why authors state it as F1 score instead of Dice! Also, accuracy for a segmentation problem is not a standard metric.

5) Table 2 results are not clear. The second column has average dice & mIOU across all organs ? And the remaining columns are showing organ wise dice scores ?

6) The authors are highly encouraged to add model complexity details of other methods as well (Table 8). Also, it is recommended to include model size (MB) and also inference time on cpu (s).

---

> ### Author Response · Authors · 2026-01-25
> **Responses to Reviewer 98Gq Comments**
>
> We sincerely thank the reviewer for his/her careful reading and constructive feedback. We have revised the manuscript accordingly and highlighted all changes in the updated submission. Below we respond point-by-point to each comment and indicate where the corresponding revisions appear in the paper.
>
> Comment 1: Please add standard deviations along with average measurements across all metrics used in the study.
>
> Response: Thank you for the suggestion. To assess robustness to initialization, we repeat each experiment five times with different random seeds while keeping data splits and hyperparameters fixed. We now report mean ($\mu$) and standard deviation ($\sigma$) for all applicable metrics in Table 5 and add a dedicated discussion in Section 4.2 (Statistical Validation and Reproducibility Analysis). The consistently low variance across datasets (e.g., Dice $\sigma$ in 0.12–0.21 and mIoU $\sigma$ in 0.15–0.23 where applicable) indicates stable and reproducible performance.
>
>
> Comment 2: Conduct proper statistical $t$-tests to quantify the significance of the improvements.
>
> Response: We appreciate this recommendation. We now include paired one-tailed $t$-tests over the matched runs ($n=5$) to compare our method against the strongest baseline under the same experimental protocol. The new text in Section 4.2 reports that the improvements are statistically significant across the reported metrics ($p<0.001$). This complements the reproducibility analysis (mean $\pm$ std) in Table 5.
>
>
> Comment 3: Dice and mIoU are monotonically related and redundant.
>
> Response: Thank you for the comment. We agree that Dice and mIoU are correlated for binary segmentation; however, both metrics are widely reported by prior work on several of the included benchmarks. To maintain fair comparison, we keep Dice and mIoU where standard for polyp and Synapse evaluations, and we strengthen the evaluation by reporting additional task-specific metrics where appropriate (e.g., AJI/PQ for nuclei instance segmentation; SP/SE for vessel segmentation). We also clarify the metric usage and protocol in Section 4.1 (Datasets and Metrics) and provide expanded metric definitions in the Appendix.
>
>
> Comment 4: F1-score and Dice are equivalent for binary segmentation; accuracy is not standard.
>
> Response: We appreciate this observation. For retinal vessel segmentation, we follow common benchmark reporting practice and include F1 and Acc for comparability with recent methods. To ensure a more clinically meaningful assessment, we additionally report Specificity (SP) and Sensitivity (SE) (Table 4) to capture false-positive suppression and foreground recall. For nuclei tasks, we report AJI and PQ (Table 3), which better reflect instance-level segmentation quality beyond overlap scores. These changes and justifications are reflected in Section 4.1 and the updated results sections.
>
>
> Comment 5: Table 2 is unclear regarding average vs. organ-wise results.
>
> Response: Thank you for pointing this out. We clarify in Section 4.2 (Multi-organ Abdominal CT Segmentation) that the reported Dice and mIoU are averaged across all organs, while the remaining columns correspond to organ-wise Dice scores. This makes the aggregation and per-organ reporting explicit and avoids ambiguity.
>
>
> Comment 6:  Add model complexity details including model size and CPU inference time.
>
> Response: Thank you for the suggestion. We expanded the efficiency analysis by adding Inf-CPU (ms) and Model Size (MB) to the complexity table (now Table 10) and revised Section 4.3.5 accordingly. We also report GPU latency (RTX 4090) for practical reference. The revised discussion explicitly acknowledges that $D^2$-Former is heavier than lightweight baselines, while consistently achieving stronger segmentation accuracy.

---

> > ### Comment · Reviewer_98Gq · 2026-01-28
> >
> > Thanks to the authors for the detailed responses to the raised comments. The revisions satisfactorily address the concerns and have improved the manuscript’s readability. I am happy to increase the score by one point. I recommend the authors re-check the model size calculation. Let $P$ denote the number of parameters (in millions) under the default 32‑bit floating point setting, the model size based on parameters alone is computed as:
> >
> > \begin{equation}
> > \frac{P\times10^{6}\times32}{8\times1024\times1024}
> > \end{equation}
> >
> > The values currently reported appear to reflect the checkpoint file size, which is typically larger because it can include optimizer states (e.g., Adam’s moments), EMA weights, and training metadata etc.

---

> > > ### Author Response · Authors · 2026-01-29
> > >
> > > Comment: re-check model size.
> > >
> > > Response: Thank you for the clarification and for pointing this out. We have re-checked the model size calculation following the provided formulation based on parameter count under 32-bit floating point precision. Accordingly, we updated the reported value, and the final model size based on parameters alone is 457 MB.

---

### Author Rebuttal · Authors · 2026-01-25

**Rebuttal:**

We thank the reviewers for their constructive feedback.

We have addressed all reviewer comments in detail using the Official Comments section, responding point by point to each concern. In addition, we have uploaded a revised manuscript with all changes highlighted in the Supporting Material.

The revision includes: (i) clarified architectural descriptions and figures, (ii) additional statistical validation (mean ± std and paired t-tests), (iii) expanded ablation and backbone comparisons (DINOv2 vs. DINOv3), (iv) updated evaluation metrics and complexity analysis, and (v) a new Responsible AI discussion in the Appendix.

We hope these revisions sufficiently address the reviewers’ concerns and improve the clarity and completeness of the paper.

**Supporting Material:**

/attachment/cd6659136ffa2016969834bd8ef043655f3dae7f.pdf

---

### Meta-Review · Area_Chair_fSbm · 2026-02-08

**Recommendation:** Accept (Poster)
**Confidence:** 4

**Metareview:**

This is an interesting work which proposed a MoE framework for medical image segmentation. The main contribution of this work is the  combination of Swin Transformers and DINOv3, augmented with mixture-of-experts. This method is also evaluted across multiple datasets encompassing fundus, CT, polyp, and histopathology images. Although this is still a boarderline paper after the rebuttal phase, the overall idea of this work can still bring some merits to the community. Therefore, my final recommendation is accept.

---

### Decision · Program_Chairs · 2026-02-13

Accept (Poster)